# The consequences of viral infection on protists
Victoria Fulgencio Queiroz[1], Juliana Miranda Tatara[2], Bruna Barbosa Botelho [1],
Rodrigo Araújo Lima Rodrigues [1], Gabriel Magno de Freitas Almeida [2,3] ✉ &
Jonatas Santos Abrahao [1,3]

Protists encompass a vast widely distributed group of organisms, surpassing the diversity observed in metazoans. Their diverse ecological niches and life forms are intriguing characteristics that render them valuable subjects for in-depth cell biology studies. Throughout history, viruses have played a pivotal role in elucidating complex cellular processes, particularly in the context of cellular responses to viral infections. In this comprehensive review, we provide an overview of the cellular alterations that are triggered in specific hosts following different viral infections and explore intricate biological interactions observed in experimental conditions using different host-pathogen groups.

The taxon "Protozoa" was formalized by the English biologist, comparative anatomist, and paleontologist Richard Owen in 1858[1]. At the time, the taxon comprised "*numerous organisms of minute size retaining the form of nucleated cells, which manifest the common organic characters, but without the distinctive superadditions of true plants or animals*"[1]. Since then, the terms protozoa, protozoans, protista and protists have been used to designate a variety of eukaryotic groups[2,3]. As this taxon does not encompass well established clades that diverged from it, protists form a paraphyletic group. Similar to other paraphyletic taxa, it proves challenging to identify universal characteristics among all its members[2]. Nevertheless, protists can be defined as predominantly unicellular eukaryotic organisms, which do not develop their tissues through the process of embryonic stratification[4,5]. Their cell morphology showcases an astonishing diversity of forms, functions, and survival strategies. They are widespread worldwide, and comprise marine, freshwater, terrestrial, symbiotic, and pathogenic strains[2,6,7]. However, taxonomists estimate that the known representatives compose only a small portion of the total of protists on Earth today, and that the diversity of this group is greater than that of metazoans[6,7].

With such abundance and richness, these eukaryotes are part of communities with a wide diversity of organisms and are especially studied for their interactions with pathogens or regarding their own pathogenicity[7]. A frequent example in literature are amoebae. Most amoebae are free-living protists of great diversity that have garnered significant attention due to their intricate associations with a plethora of microorganisms from all domains of life[8–19]. Free-living amoebae occupy different ecological niches and have already been found in a wide range of natural and anthropized environments[7]. The most notorious feature

of amoebae is their ability to alter their cell shape by creating temporary extensions of cytoplasm known as pseudopods, which serves both for feeding and movement[7]. Due to their foraging behaviors and predatory nature, amoebae are often considered as professional phagocytes, mainly consuming microorganisms to fulfill their nutritional needs[9]. However, many microorganisms successfully evade the phagocytic pathway and thrive within the amoebae, turning them into transmission vehicles or incubators and enabling various types of ecological relationships to occur, from symbiotic to parasitic[9]. Besides, their cell biology and active grazing behavior serve as a widely used infection route for giant viruses, although miscellaneous entry mechanisms have already been described for this phylum of viruses[10,20,21]. Given the substantial size of their viral particles, they can trigger phagocytosis in amoebae, creating entry opportunities[11–14]. In fact, the efficiency of this route lead other giant viruses that lack the requisite size for phagocytosis to employ a mimicry strategy to use the phagocytic pathway as an entry route[15].

The myriad of interactions involving amoebae renders them exceptional experimental models to explore and investigate a diversity of scientific fields[16]. Over the years, amoebae have contributed to the understanding of a wide array of subjects. These encompass the unraveling of mechanisms related to the resistance and pathogenicity of microorganisms, the intricacies of cell locomotion, the functioning of non-muscle contractile systems, the dynamics of populations and communities, the implications of cell nucleus removal and transplantation, events involving horizontal gene transfer, and the evolution of organelles, that may provide valuable insights into the origin and evolution of eukaryotic cells[9,22–29]. This shows that the study of protists can help advance different fields of science. However, to

[1]Federal University of Minas Gerais, Institute of Biological Sciences, Department of Microbiology, Belo Horizonte, Minas Gerais, Brazil. [2]The Norwegian College of Fishery Science, Faculty of Biosciences, Fisheries and Economics, UiT - The Arctic University of Norway, Tromsø, Norway. [3]These authors contributed equally: Gabriel Magno de Freitas Almeida, Jonatas Santos Abrahao. ✉e-mail: gabriel.d.almeida@uit.no

gain a more comprehensive understanding, it is important that we delve deeper into the basic aspects of protist cell biology.

## Viruses are unique tools to study protists biology

Viruses have been unique tools in helping to comprehend complex biological processes in different host models. As mandatory intracellular parasites, viruses have an intimate relationship with their host cells, often depending on and controlling their cell structure, metabolism, biochemical machinery and behavior[30,31]. The diversity of structures, genomes, and replication strategies that viruses exhibit reflects thousands-to-billions of years of coevolution with their hosts, some older than the origin of the eukaryotic cell[32]. This is a key point in biology, as over the centuries, the study of host-virus interactions has led to important discoveries. For instance, in genetics, these findings include the discovery of DNA as the source of heritage, mRNA, mRNA processing, RNA interference, as well as the regulation of gene expression, transcriptional control elements, and

**Fig. 1 | Overview of the current extent of the known protist virosphere. a** Protist supergroups are shown on the *x*-axis while viral groups are shown on the *y*-axis. Highlighted squares on the matrix represent a group of viruses infecting a respective group of protists. The clades Viridiplantae, Fungi and Metazoa were removed from the dendrogram (latest eukaryotic phylogeny was retrieved and adapted from Keeling & Eglit, 2023[223]. Viral groups at family or genus level (unclassified cressdnavirus and preplasmivirus at phylum level and unclassified imitervirus and picornavirus at order level) are also on the *y*-axis. **b** The viral counts at genus level for each virus class (DNA or RNA, single-stranded or double-stranded) were retrieved from VirusHostdb[224] and manual curation of the literature. Abbreviatures: double stranded (ds), single stranded (ss), negative sense (−), positive sense (+), double-stranded DNA genome that has an RNA intermediate (dsDNA-RT). Detailed information regarding viruses and respective hosts used to make this figure can be found in supplementary data file (tab 1).

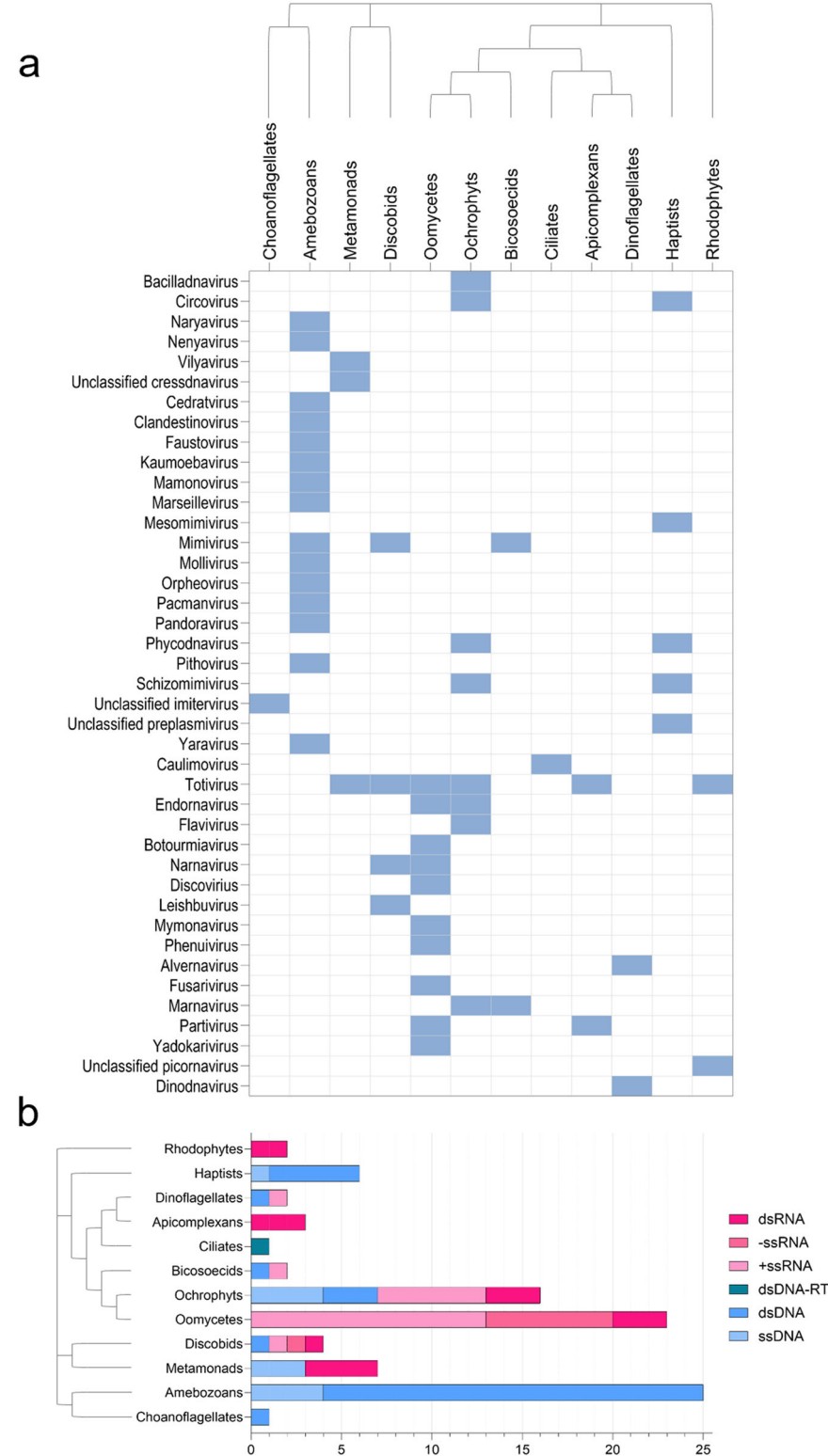

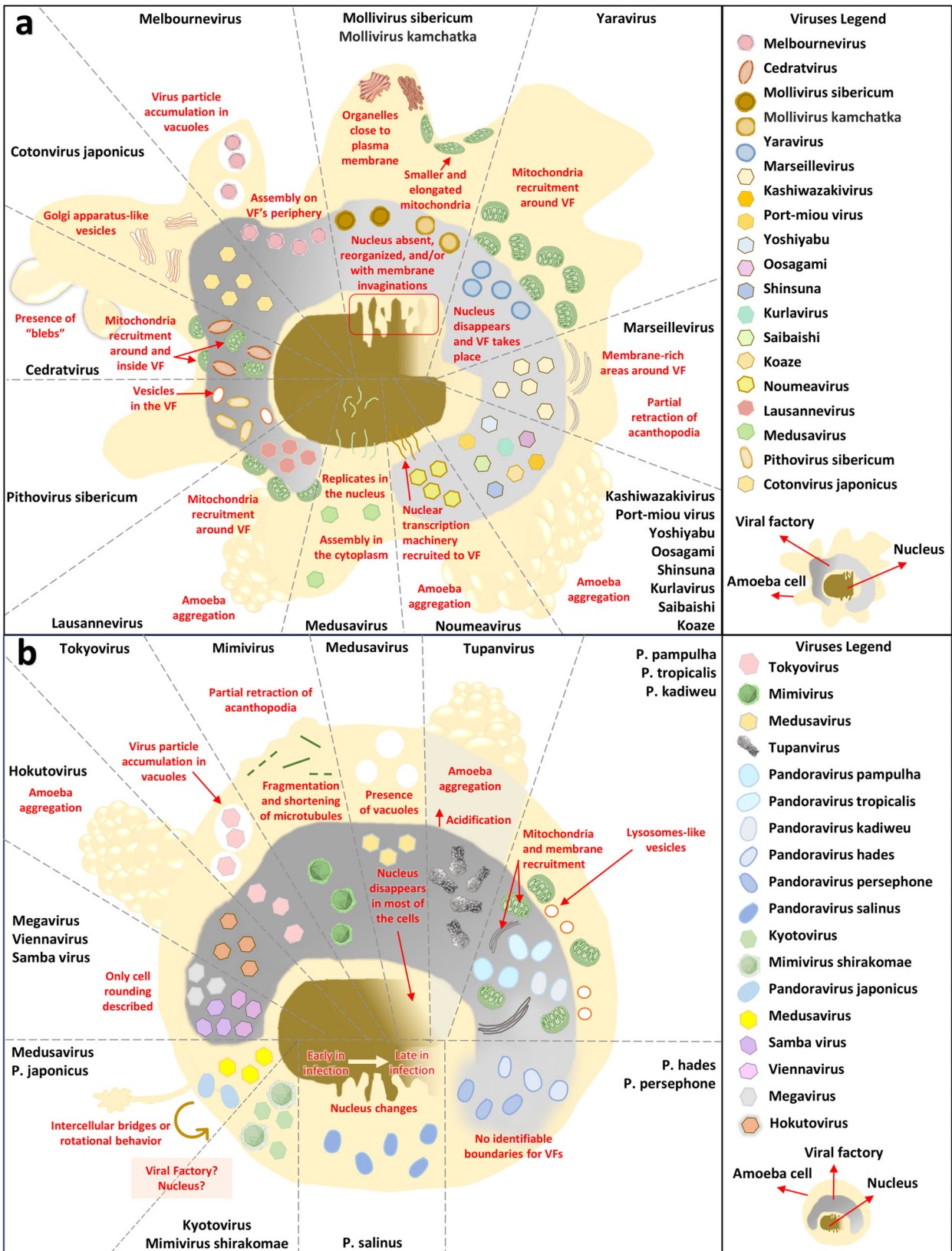

**Fig. 2 | Overview of the main changes in the *A. castellanii*. cell during different giant viruses' infection. a** "Trophozoite cell shape" to represent viruses that did not interfere in the cell shape structure. **b** "Rounded cell shape" to represent only the viruses reported to trigger cell rounding. Red text: cell modifications.

transcription factors[30,31,33,34]. Additionally, these studies have contributed significantly to our comprehension of biochemistry, elucidating tyrosine kinases and signal transduction pathways[30,31,33]. Furthermore, they have shed light on immunology and many other intrinsic aspects of host cells. These include our understanding of histocompatibility antigen function, interferons, cellular oncogenes, tumor suppressor proteins, and apoptosis in eukaryotic hosts[30,31,33,35]. They've also contributed to our knowledge of the CRISPR/Cas9 mechanism in bacterial and archaeal hosts, which is now widely used for genome editing for many biotechnological purposes[36].

Undoubtedly, the field of virology has made substantial contributions to science, extending beyond its significance in diseases. Throughout history, the discovery of viruses has primarily been driven by the concerns of human health, and the health of other animals and plants of economic importance[30,33,37]. This implies that most of the described viruses have direct connections to humans, whether in the realms of economics, medicine, or biotechnology, resulting in an extremely anthropocentric known virosphere[37]. However, the viruses known to humankind represent only a small fraction of the viral diversity on Earth. As viruses infect organisms of the three domains of life, and some studies estimate the count of viral particles on Earth to be of the order of $10^{31}$, there remains a large number of undiscovered viruses[35,38]. Still, due to the absence of a universal molecular marker for viruses, the true extent of global viral diversity remains largely unknown.

Despite having a small number of identified representatives so far, viruses that infect protists exhibit a remarkable diversity (Fig. 1). One of the earliest mentions of virus-like particles (VLP) in protozoa seems to have been documented in 1960 within *Entamoeba histolytica*[39]. Over the years, many taxa of viruses that infect protists have been identified, with representatives containing DNA or RNA genomes of various topologies and sizes, which can exist either in a naked form or be enclosed within capsids, that also exhibit diverse sizes and symmetries[40–47]. Among these viruses, a group that garners significant attention due to the complexity of their genomes and the size of their particles are the viruses of the phylum *Nucleocytoviricota*. Regardless of their hypothetical monophyletic origin, the viruses belonging to this phylum display a broad host range within eukaryotes. This group includes the giant viruses (GVs) that infect amoebae, the ones holding the distinction of possessing the largest genomes and viral particles found in the virosphere[48]. Although these viruses infect amoebae in the laboratory, their natural hosts are still uncertain. Over the past two decades, there has been dedicated research into the exceptional biological traits of these viruses, the enigmatic composition of their genomes and particles, and their ecological significance.

Virology is an integrative science. Host-parasite interactions can reveal valuable information about cellular alterations and host defense mechanisms that can be used to study basic cellular processes[30,33]. During a viral infection, the virion and host experience modifications of such magnitude that both start to present a different nature, sometimes even shifting the cellular role to solely producing the viral progeny. In consideration of this, the term 'virocell' was coined to refer to this specific moment when a virus is actively infecting a cell[49]. One of the most characteristic changes during a viral infection is the establishment of a viral factory (VF), that can be formed in both the cytoplasm or nucleus of the cell. It is the place where viral progeny is assembled, and often, the genetic material is replicated. Its formation is usually accompanied by extensive cytoplasmic rearrangements[50]. Therefore, studying host-pathogen interactions can uncover how protists respond to viral infections and highlight the features that were evolutionarily selected to counter viral threats. Moreover, viruses can be engineered for various biotechnological purposes, including gene delivery and genome editing. This technology can help elucidate protist biology for basic research or practical purposes[51]. Besides, these studies can also provide enlightenment into general cell biology that may apply to a wider range of eukaryotic organisms. In this review, we provide an overview of the effects of virus infection and replication on protists and discuss some prospects that the investigation into viral infections in protists could offer.

## Cell machinery and morphology modulation in protist cells during viral infections

Over the past two decades, studies concerning giant viruses have highlighted virus-protist interactions[52]. Amoeba, particularly the genus *Acanthamoeba* and the species *Vermamoeba vermiformis*, are commonly studied hosts that suffer cell machinery and morphological changes after infection[53]. The ciliate *Tetrahymena* sp. also shows alterations following GV inoculation. Limited details exist for protists infected with GVs and other viruses, including diatoms, flagellates, dinoflagellates and algae. A list of publications mentioning cellular alterations derived from viral infections in protists is shown in the supplementary data file (tab 2). Below, we discuss these alterations in detail for each main host group.

### *Acanthamoeba castellanii* and their broad response to different viruses

Most known GVs groups were isolated in *A. castellanii*[54]. A successful giant virus infection triggers the formation of VFs inside the host cells. However, some studies mention cytopathic effects (CPEs) without evident VFs[55–57]. Nevertheless, GVs' DNA replication may also happen in the nucleus, with assembly in cytoplasm[58]. Noumeavirus, for instance, recruits the host's nuclear transcription machinery to the cytoplasmatic VF[59]. An overview of some of the main changes using the *A. castellanii* model as an example can be seen in Fig. 2.

Some GVs infections can disorganize and deform the nucleus of *A. castellanii*, leading to loss of its spherical appearance, membrane invaginations, or degradation[55,60–66]. The cytoplasm may undergo significant changes including acidification, organelle rearrangement, cytoskeleton modulation with fragmentation and shortening of microtubules, lower quantity of vacuoles, viral particles present inside vacuoles, formation of Golgi apparatus-like vesicles, among others[59,61,63,64,67–72] (Fig. 2a).

Viral infection can also modulate the energy of the host, redirecting it to the replication site, often involving the recruitment of mitochondria to VFs[60,65,73,74]. Mollivirus sibericum infection results in smaller and elongated mitochondria, although host cells did not present changes in cellular adhesion and morphology[62]. Despite that, cell rounding is one common CPE, observed after GVs infection (Fig. 2b)[75–78]. Studies also reported loss of cell adhesion decreased motility, and intercellular bridges or rotational behavior[55,56,76,79,80]. A peculiar CPE is cellular aggregation ("bunches")[77,81,82]. This mechanism was suggested to increase viral dissemination during Tupanvirus infection, as mannose-binding protein gene transcripts are significantly increased at earlier times of infection (1, 2 and 4 h post-infection [HPI]) and precedes the formation of bunches (6 HPI) between infected and uninfected cells.

### *A. polyphaga* and *V. vermiformis* morphology modifications upon giant viruses' infection

Fewer studies described CPE in *A. polyphaga* during GVs infection. Acanthamoeba polyphaga mimivirus (APMV) infection leads to nucleus enlargement, prominent VF formation in the cytoplasm, multivesicular bodies and endoplasmic reticulum (ER) membranes appearance around VF, stimulus of ER synthesis, significant structural alteration in microtubules and actin microfilaments resulting in cells becoming smooth, rounded, and losing motility[83–85]. Marseillevirus infection induces similar VFs to the observed in *A. castellanii*, with lack of a clear boundary with cytoplasm[67,86]. Furthermore, cell rounding with formation of cell chains was reported for the first time after a viral infection in amoeba[87].

Over the past decade, studies implemented *V. vermiformis* as a new isolation platform, leading to the discovery of new GVs[12,13,88–94]. Their infection resulted in cell rounding and loss of adhesion[88,94], and cell stretching with increased motility before cell rounding[12]. The formation of "blebs" and cell bunches seen in *V. vermiformis*, were previously reported in *A. castellanii*[12,73,91]. A distinct response was the encystment of neighbor non-infected cells after the exposure to infected cell-released non-proteic soluble factors, serving as a defense mechanism, as those cysts become unviable

**Table 1 | Overview of the proteomic and transcriptomic studies of different protist organisms infected by their respective viruses**

| References | Protist organism | Virus | Technique | Time post infection | FoldChange | Down | Up |
|---|---|---|---|---|---|---|---|
| Rodrigues et al. (2020)[137] | *A. castellanii* | Marseillevirus | RNAseq | 0, 1, 2, 4, 5, 6, 8, 10, and 12HPI | <−1 and >1 | 28 | 19 |
| Legendre et al. (2015)[62] | *A. castellanii* | Mollivirus sibericum | LC-MS/MS | 0 to 6HPI | <−2 and >2 | 38 | 30 |
| Zhang et al. (2021)[138] | *A. castellanii* | Medusavirus | RNAseq | 8 to 16HPI | <−0.445 and >0.364 | 7970 | 2657 |
| Moniruzzaman et al. (2018)[160] | *A. anophagefferens* | Aureococcus anophagefferens virus | RNAseq | 5 min | <−1.5 and >1.5 | 588 | 412 |
| | | | | 30 min | | ≈ 865 | ≈ 505 |
| | | | | 1HPI | | 82 | 0 |
| | | | | 6HPI | | ≈ 1060 | ≈ 690 |
| | | | | 12HPI | | ≈ 1260 | ≈ 1260 |
| | | | | 21HPI | | ≈ 1430 | ≈ 1620 |
| Poimala et al. (2022)[138] | *Phytophthora cactorum* | PcBV1 & 2 | RNAseq | -a | <−2 and >2 | 23 | 10 |
| | | | Nanoflow reverse-phase LC-MS | -a | <−1.19 and >1.37 | 17 | 36 |
| Provenzano et al. (1997)[171] | *T. vaginalis* | TVV | 2D | - | - | 41 | 47 |
| He et al. (2017)[174] | *T. vaginalis* | TVV | iTRAQ labeling | - | <−1 and >1 | 21 | 29 |
| Rada et al. (2022)[175] | *T. vaginalis* (exossome vesicle) | TVV | LFQ-MS | - | <−2 and >2 | 55 | 20 |

"FoldChange" means threshold utilized by each study to determine when a protein or transcript presents increased expression (upregulated - "up") or decreased expression (downregulated - "down").
Other protists are not included on this table since information on the number of deregulated proteins or transcripts was not the focus of the publication or this data was not provided.
aThe virus was removed from the protist strain, thus there was no time post-infection.

trapping the virus inside[90]. Nuclear alterations of infected *V. vermiformis* include loss of the rounded shape, decrease in surface area, or nucleus disappearance[12,13,90–94]. Furthermore, clandestinovirus formed its VF within the nucleus[88]. *V. vermiformis'* mitochondrias were also reported to be recruited around VFs[12,13,90,91,93,94]. Investigating the unique cellular responses in amoebae to various GVs is essential for establishing correlations in host-virus interaction and comparing potential CPEs to other types of host cells.

## Other protists' morphology and behavior in response to virus inoculation

When exploring protist susceptibility to Tupanvirus beyond amoebae, it was found that the ciliate *Tetrahymena* sp. experience cytotoxic effects in response to the viral presence. The morphological changes included gradual vacuolization, nuclear degradation, loss of motility, and formation of vesicles, similarly to the observed in *A. castellanii*. However, the cilliate does not support viral replication[61].

In contrast to the extensive use of amoeba to study viral infections, there are relatively fewer reports focusing on interactions between viruses and other protists. Among algae protists, the viral infection of *Aureococcus anophagefferens*, a component of ocean blooms, can trigger nucleus and organelle degradation (such as chloroplast) in some infected cells, cell lysis, and loss of the outer polysaccharide glycocalyx during viral infection, even in environmental samples[95–99]. In addition, viral infection of another bloom-forming protist, *Heterosigma akashiwo*, pointed to nucleus degradation and damaged cell wall[100]. When infected by HaNIV virus, cell nucleus presented margination of heterochromatin, but no morphological changes[101]. Conversely, the infection by HaV01 and HaV triggered cell rounding, and loss of cell motility, together with chloroplast degradation for HaV infection[102,103]. Interestingly, chloroplasts have been reported to remain intact until cell lysis[104]. Vacuolation, disintegrated cytoplasm and ER swelling were reported during HaRNAV infection in *H. akashiwo*[105]. Furthermore, in the haptophyte algae *Chrysochromulina parva* and *Prymnesium parvum* the infection by their respective viruses resulted in absence or degradation of cell nucleus[106,107].

In the diatoms *Chaetoceros* sp, *Guinardia delicatula* and *Rhizosolenia setigera* the infection by their respective viruses resulted in common CPEs: degradation of chloroplasts and/or photosynthetic pigments[98,108–113]. Few *Chaetoceros* species additionally exhibited cytoplasm degradation[108–111,114], and nucleus or nucleolus degradation was reported in *Chaetoceros salsugineum* and *Guinardia delicatula*[98,115]. Similarly, chloroplast shrinkage and chlorophyll degradation were suggested during the infection of the phytoplankton coccolithophore *E. huxleyi* by its lytic virus[116,117]. Nucleus degradation was identified in infected *E. huxleyi* cells, as well as cellular aggregation, likely as a defense strategy to sink infected aggregated cells[116–118]. Furthermore, mitochondrial damage with vacuolar acidification was described during the infection byf EhV99B1[119].

Regarding flagellates, the protist *Giardia* sp. has two life cycle stages, a trophozoite and a cyst form[120,121]. Only its trophozoite stage is susceptible to viral infection, and few morphological changes have been reported for *Giardia canis* and *Giardia lamblia* infected by GCV and GLV, respectively. During GCV infection, the protists presented vacuolization, enlargement of ER, and its cytoplasm became very loose, and during GLV only cell adherence was impaired upon infection[122,123]. Although, no cell death was reported for both studies, corroborating to the suggestion that giardiaviruses are released from the cell without triggering cell lysis[121]. *G. duodenalis*, however, experiences reduction in cell growth without apparent morphological changes[124]. In the free-living flagellate *Bodo saltans* infected by the virus BsV, morphological changes included degraded nucleus and intracellular structures, with lipid vesicles migrating through the VF, while kinetoplast remained intact for longer[96].

The bloom-forming dinoflagellate *Heterocapsa circularisquama*, associated with red tides and bivalve mortality, can be infected by RNA and DNA viruses. During the DNA virus infection, a VF is formed with granular or fibrous material inside, in addition to organelles disruption[125–127]. However, the RNA virus infection triggered organelles, nuclear and chlorophyll-a degradation, and loss of cellular motility[128–130]. Interestingly, some of the cells were resistant to the RNA virus infection, but not the DNA one, and the non-resistant cells underwent cell lysis[129,131]. Furthermore, the dsDNA virus

infection triggered damage to *Symbiodinium* cell nucleus initially, and later to organelles[132]. The dinoflagellate *Gymnodinium mikimotoi* shows nuclear degradation and swollen chloroplasts upon viral infection[133]. Nuclear degradation is also reported to the nonphotosynthetic stramenopile from the genus *Sicyoidochytrium* upon viral infection[134]. Overall, relating viral responses across different protist organisms and viruses remains challenging, due to the limited number of studies in this field and the scarcity of available isolates for research.

## Metabolic changes in protist cells during virus infection

To investigate cell-virus interactions at the molecular level, different approaches, including transcriptomics and proteomics, have been employed, revealing that each virus modulates their host in a unique manner[135,136].

### Changes in *Acanthamoeba* sp

Proteomic analysis of *A. castellanii* infected by Mollivirus sibericum identified a few upregulated proteins related to histones, autophagy, DNA synthesis and packaging; and a few downregulated proteins with no clear functional relationship[62]. Together with the fact that this virus is released through exocytosis rather than cell lysis, an overall cellular integrity is suggested throughout the replication cycle[62]. Likewise, few genes were deregulated at the transcriptional level during Marseillevirus infection[137]. Transcripts expression across early, intermediate, and late time periods showed only 48 genes deregulated among these periods (Table 1), with the upregulation of proteins mainly linked to exosome secretion, transfer RNA biogenesis and genetic information processing; and the downregulation of proteins linked to translation apparatus (rRNA related proteins), tRNA encoding genes, carbohydrate metabolism and lipid biosynthesis[137]. For Marseillevirus, amoebal and mitochondrial transcript levels decreased after 1 h post-infection (HPI)[137].

Contrastingly, Medusavirus infection in *A. castellanii* resulted in thousands of differentially expressed host genes, although mitochondrial gene modulation resembled the pattern of Marseillevirus infection. Only 25% of nuclear host genes, mainly related to ribosome and proteosome pathways, increased its expression between 8 to 16 HPI, suggesting the host to be suffering protein degradation, along with increased viral protein synthesis. The remaining 75% experienced decreased expression during that same period. Gene ontology suggested a reduction in transport activity, although at 48HPI many of these genes were downregulated, in addition to an upregulation of encystment-mediated genes[138].

Furthermore, Tupanvirus inoculation triggered rRNA shutdown only by the presence of viral particles, unrelated to viral replication[61]. Genomic analysis of tupanviruses revealed the involvement of the ribonuclease T2, an enzyme related to the reduction of the physiological activity and phagocytosis capacity in protist hosts[139].

### Bloom-forming and dinoflagellate protists

Infected by lytic and non-lytic viruses, the coccolithophore *E. huxleyi*, is ecologically crucial for marine carbon flux[140,141]. The correlation between transcriptomics and metabolomics analysis during viral infection indicated that early during infection (1–4 HPI), there was an upregulation of sphingolipid biosynthesis and glycolysis shuffling energy to fatty acids biosynthesis. Later (24HPI), a shift to the activation of the pentose phosphate pathway to produce nucleotides occurred, while glycolysis and fatty acids became downregulated[117]. Similar results were found for genes related to sphingolipid metabolism, such as the upregulation of dihydroceramide desaturase (DCD) at 6HPI and its downregulation at 45HPI[142]. *E. huxleyi* DCD transcript, together with serine palmitoyl transferase (SPT) transcript (another enzyme from sphingolipid biosynthesis pathway) level results pointed to the same tendency of a decrease in these host transcripts production in the environment[143]. Other studies also found lipid modulation in *E. huxleyi* during viral infection, including fatty acids and highly saturated triacylglycerols, and lipid metabolism was suggested to be regulated by modulation of the PI3K-Akt-TOR signaling pathway[144–152].

Furthermore, a single cell transcriptomics analysis of *E. huxleyi* during infection revealed an early shutdown of nuclear transcripts, with mitochondrial and chloroplast transcripts initially higher but gradually declined[153]. Indeed, there was a decrease in genes involved in photosynthesis at 6, 12 and 24 HPI, although some cells seemed to be intact and photosynthesizing[148]. Enriched functions associated with *E. huxleyi* responding to viral infection included modified amino acid, lipid binding, porin activity, calcium channel activity, pore complex, cell outer membrane, bacterial-type flagellum functions, among others[154]. High expression of glycolysis and nucleotides biosynthesis related genes were reported when studying single protist cells from a coccolithophore bloom[151]. In addition, there was an upregulation of autophagy related genes, with decreased expression of negative regulatory factors, such as PI3K, and an increase of reactive oxygen species (ROS), which is all related to the programmed cell death (PCD) triggered in *E. huxleyi* after infection[119]. In fact, 20 ROS scavenging genes were impacted after viral infection, and ROS-related genes were found to be increased through transcriptomic analysis[155,156]. Cell death induced by viral presence could occur before viral particles release, which can be associated with cell autophagy as a defense strategy to avoid viral dissemination, although autophagy is also essential for viral propagation[117,157]. *E. huxleyi* cell cycle can be affected by virus infection through the modulation of cyclin expression, and host life cycle genes have been found to be impacted during viral takeover of the protist in environment[149,158,159]. Altogether, these findings suggest substantial cellular changes in *E. huxleyi* after viral encounter, influencing factors related to cell death, modulation of energy, and specific lipids production, alongside a decline in nuclear activity, potentially playing a role in the modulation of oceans blooms[153].

Another bloom forming protist, *Aureococcus anophagefferens*, exhibited transcripts downregulated at six different time points during viral infection. A massive deregulation started as early as 5 min post-infection and continued after 6HPI. Downregulated proteins indicated the suppression of pathways related to host cytoskeleton formation, photosynthesis, fatty acid metabolism, and carbohydrate biosynthesis. The upregulated group of proteins indicated the activation of host cellular respiration, transcription, protein synthesis, polyamine biosynthesis, and RNA processing. Transcripts related to host defense mechanism were likely suppressed. This dramatic cell modulation indicates the virus –host interaction in a time dependent manner until cell lysis occurs, correlating with the virus's role in brown tide blooms[160].

For *Phaeocystis globosa*, viral infection prevented the accumulation of polyunsaturated fatty acids, decreased protist photosynthetic performance and upregulated mitochondrial respiration, triggered the fragmentation of DNA and activation of caspases, and prevented *P. globosa* to release star-like structures, which in turn affects host carbon assimilation[161–164]. The correlation among such changes is speculative, suggesting an association with the lysis mechanism triggered in this organism post-viral infection.

Comparing the transcriptomic responses of the dinoflagellates *Symbiodinium tridacnidorum* and *Symbiodinium* C3 under or not UV light exposure (with and without viral replication, respectively), both exhibited upregulated viral transcription and related terms, but only *S. tridacnidorum* triggered downregulation of transcripts related to host-response to viral infection[165].

### Protists responsible for causing human or plant disease

After the discovery of Trichomonas vaginalis virus (TVV), few studies investigated *Trichomonas vaginalis* cell responses to infection[166,167]. Initial findings described that most clinical isolates had the presence of a dsRNA icosahedral virus, and that the expression of a major immunogen (P270) on cell surface, and the expression of Ig-degrading proteinases, were directly correlated with its presence[168–170]. An overview of the protein expression pattern of *T. vaginalis* after TVV infection found at least 47 expressed and 41 suppressed proteins linked to the virus[171]. Isolates of *T. vaginalis* might be infected by up to 4 different strains of TVV, and only the cells infected by TVV2 and TVV3 are capable of inducing P270 protein expression[172,173].

Proteomic studies comparing infected and uninfected cells detected the upregulation of adhesin proteins related to the pathogenicity of *T. vaginalis* when infecting humans[174]. Further, metabolic enzymes, ribosomal and heat shock proteins were also differentially impacted[175]. Extracellular vesicles of *T. vaginalis* infected by TVV exhibit differences in protein content, such as a protein responsible for increasing adherence, suggesting enhanced exosome binding and pathogenicity[175]. Overall, TVV plays an important role in *T. vaginalis* dynamics by modulating different genes that aid the protist to successfully establish in its host.

After the discovery of endosymbiotic viruses in *Leishmania*'s cytoplasm, most studies focused on the relation *Leishmania*'s pathogenicity to humans when infected or not by one of its two viruses, LRV and LBV[176–178]. The metastasizing *L. Viannia* had a higher amount of LRV1, suggesting that the virus triggers parasitic resistance[179]. A study looked for transcriptional changes in *L. guyanensis* and *L. major* when removing their respective viruses LRV1 and LRV2[180]. As a result, *L. guyanensis* did not suffer significant changes, with only 2 differentially expressed genes. On the other hand, proliferation rate of *L. major* cells decreased, with 67 upregulated and 20 downregulated transcripts after its virus removal. Among the downregulated group, there was a cyclin related to growth kinetics; and the membrane-bound acid phosphatase 2 enzyme, which has been already suggested to affect the biology of *Leishmania*. The upregulated group presented transcripts related to autophagy, cell response to various stimuli, and nucleosome assembly[180].

Oomycetes from the genus *Phytophthora* are responsible for causing great damage to agriculture[181]. A metagenomic study of *Phytophthora condilina* resulted in 15 different putative viruses identified[182]. Moreover, GVs sequences have been identified in the genome of *Phytophthora parasitica*[183]. Phytophthora endornavirus 2 (PEV2) and Phytophthora endornavirus 3 (PEV3), are suggested to stimulate zoosporangium development and inhibit hyphal growth, also reducing the host oomycete sensitivity to the antifungal metalaxyl[184]. In addition, seven different viruses were able to replicate in *P. infestans*, and viral infection apparently did not affect the cell host morphology but stimulated the growth of the mycelium mass[185]. Notably, the infection of *P. infestans* by PiRV-2 is suggested to increase the protist sporulation, which in turn is associated with a likely hypervirulent factor against its plant host[186]. However, sporulation of *P. cactorum* infected by PcBV1 & 2, seemed deeply affected when compared to *P. cactorum* with PcBV1 & 2 ablation. Its sporangia production and size decreased during viral infection, together with reduced hyphal growth[187]. This same research used *P. cactorum* as a model to explore the transcriptomic and proteomic changes after removing PcBV1 & 2 from the protist, and identified 10 up- and 23 downregulated transcripts, as well as 36 up- and 17 downregulated proteins[187]. The excreted protein elicitin was found upregulated in the infected cells[187]. This protein is related to the suppression of the plant immune response but can also be recognized by the plant and become a factor that reduces protist pathogenicity[188,189]. Overall, *Phytophthora* can be pathogenic or not to its plant host depending on many factors, including the infecting virus type.

## Cellular resistance strategies after stress exposure

All living organisms are subjected to the exposure of extrinsic factors that can trigger a stress response at the cellular level. Stress refers to any environmental condition that can be harmful to cells and induce physiological changes that disturb homeostasis[190]. Biotic stress is caused by interaction with other living beings that will act as stressors, particularly parasitic relationships involving viruses, bacteria, and fungi. Abiotic stress arises from physicochemical factors that can affect cell physiology, such as pH, temperature, radiation, osmolarity and chemical molecules[190,191]. Despite its disruption of homeostasis, environmental stress is an important factor when it comes to evolution. Cells capable of restoring homeostasis after stress, called acclimatized cells, are favored through natural selection[190]. Protist cells employ various responses to stressors programmed cell death (PCD). One example is the response of *Peridinium gatunense* to oxidative stress. At the end of the algal blooms caused by this dinoflagellate, $CO_2$ becomes

significantly limited, leading to the production of reactive oxygen species which trigger PCD[192]. *Aureococcus anophagefferens*, responds to stress conditions such as low concentrations of inorganic nitrogen and inorganic phosphorus through a transcriptional shift: transcripts related to nitrogen transport and metabolism, and transcripts encoding enzymes that hydrolyze organic phosphorus or alleviate arsenic toxicity are upregulated in these scenarios. In addition, in the context of low light levels, which acts as a stress factor, transcripts encoding enzymes that catabolize organic compounds, restructure lipid membranes, or are involved in the biosynthesis of sulfolipids are upregulated to restructure lipids and renovate the photosynthetic apparatus. The cell undergoes several physiological changes to maintain the survival and ecological success of the species[193].

## Spore formation as a response to infection

The marine diatom *Chaetoceros socialis* responds to stress caused by adverse conditions of temperature, light and lack of nutrients through spore formation. Viruses are widely distributed and persistent biological stressors, and some of them are also capable of inducing spore formation[194]. For instance, CsfrRNAV01 can effectively induce significant spore formation in *C. socialis*. Viral infection acts as a biotic trigger that induces a substantial formation of heavily silicified spores. Interestingly, their spores do not produce infectious viral particles, making this shift in the life cycle an effective defense strategy against viruses and preventing the loss of a portion of the protist population[194].

### *Emiliania huxleyi* life cycle shift as a defense strategy

Another protist that relies on a life cycle shift is *E. huxleyi*, a photosynthetic organism widely distributed throughout the oceans. It has two independent phases, haploid and diploid, each with distinct morphologies[195]. The haploid form (N) consists of biflagellated cells covered by thin, organic, nonmineralized scales. In contrast, the diploid (2 N) is nonmotile and contains minute calcite platelets, named coccoliths[196]. The latter is responsible for large natural algal blooms, an important environmental problem[197]. It is known that calcium plays an important role during the viral entry process into the host cell in different infection models[198–201]. One significant difference between the N and 2 N phases of *E. huxleyi* is the calcium metabolism being considerably greater in the diploid phase than in the haploid phase, affecting virus-host interactions[197]. Haploid cells are resistant to infection and subsequent lysis by Emiliania huxleyi virus, whereas this infection contributes to the decline of diploid populations during algal blooms[197]. Evidence suggests that those lytic viruses are responsible for demising the algae population and terminating algae blooms. Thus, Emiliania huxleyi viruses are responsible for regulating the population of this coccolithophore both in abundance of cells in the environment and in composition[195,197,202,203]. Such protist-virus interaction can directly activate the life cycle transition, since the oxidative stress caused by viral infection can trigger the diploid-to-haploid shift[197]. Viral infections are known to increase the production of reactive oxygen species in 2 N cells, which induces the activation of metacaspases and, thus PCD is triggered[204]. However, oxidative stress response leads not only to cell death but also to induction of the aforementioned shift[196]. Although studies conducted using 1 N *E. huxleyi* strains found viral RNA and small amounts of EhV glycosphingolipids within the cells[146,205], a later study using the host resistant strains have not found any EhV genetic material, which suggests that there is no resistant strain bearing any form of the virus[195]. Haploid virus-resistant cells are produced as a response to viral infection[195]. This is referred to as the 'Cheshire Cat' strategy, in which the organism escapes the parasite by shifting to a life stage that is infection-resistant[197].

### Encystment as amoebae' resistance form against viral infections

The 'Cheshire Cat' theory extends to describe a relationship between a giant virus and an amoeba as well. Some amoebae hosts are also capable of shifting to an infection-resistant life form[206]. Amoebae have two life cycle stages, trophozoites and cysts. The trophozoite stage is the one that predominates when environmental conditions are favorable, such as nutrient supply and

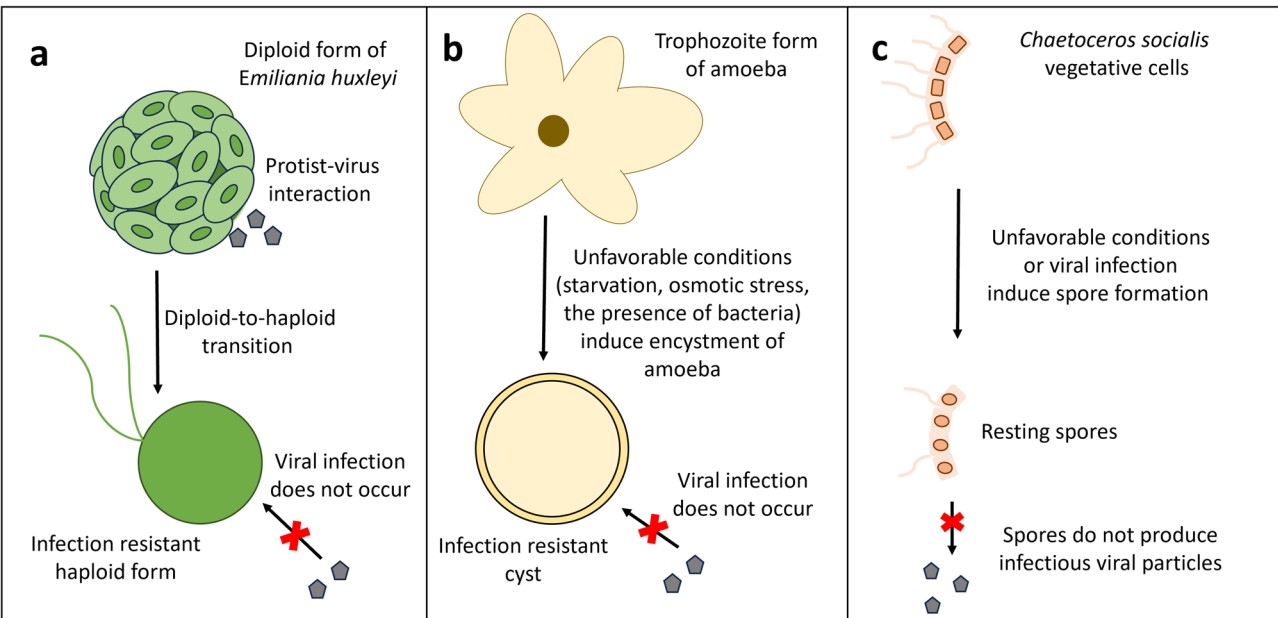

**Fig. 3 | Overview of the strategies employed by different protist life forms to resist stress factors, such as virus infection. a** Upon interaction with viruses, the coco-lithophore *E. huxleyi* shifts to a life form that is resistant to infection[195]. **b** Amoebae encyst under unfavorable conditions, and the cysts formed are not able to be infected by a virus[206]. **c** Under unfavorable conditions *Chaetoceros socialis* cells transform into resting spores that are unable to produce infectious viral particles[114,194,225]. Red text: Cellular resistance strategies after stress exposure.

temperature, while the cyst is the form of resistance[191,206]. Under unfavorable conditions such as starvation, or osmotic stress, amoeba trophozoites undergo both cellular and molecular regulatory processes that lead to encystment[206–208]. Amoeba cysts are more resistant to adverse conditions. For instance, *A. castellanii* cysts can survive five cycles of freeze–thawing, exposure to high doses of UV or gamma radiation[209]. Furthermore, giant viruses, such as mimiviruses and marseilleviruses, can only infect amoeba when they are in the trophozoite form. The shift to a non-permissive cell stage is an astounding stratagem for escaping giant viruses[206]. Some viruses, however, have developed strategies to circumvent this resistance mechanism. Mimivirus isolates and Tupanvirus, can prevent the encystment of amoebae, by reducing the expression of amoebal encystment-mediating subtilisin-like serine proteinase, a proteinase involved in trophozoite encystment. By doing so, the virus can effectively block the formation of the host's form of resistance and continue its cycle[90,206,210].

## The tripartite systems
The tripartite systems are an interesting dynamic in which the giant virus-host relationship is accompanied by a further member: a virophage or a bacteria. Such tripartite systems are found, for instance in amoebae. Virophages are viruses capable of infecting giant viruses, they multiply inside the VFs, meaning that they do not replicate if giant viruses are absent. Therefore, the replication of the virophages interferes with the cycle of the giant virus, interrupting it and thus reducing the amoeba mortality related to the viral infection[79,211]. The first virophage discovered was sputnik, a dsDNA virus associated with mamavirus. Some types of virophages increase the formation of abnormal giant virus particles, reduce infectivity and the ability to lyse their host cell[211]. The second virophage discovered was the mavirus associated with the Cafeteria roenbergensis virus (CroV). Like the sputnik, this virophage inhibits the cycle of the giant virus. Interestingly, mavirus can integrate its genome into the protist's genome, but its genes are only expressed when CroV infection occurs. This reactivation is not enough to prevent cell lysis; however, it causes mavirus particles to be released into the environment and protect neighboring cells[79,212,213]. Despite this, it has been shown that not all virophages negatively affect the cycle of giant viruses; zamilon, for instance, does not affect the ability of the giant virus to lyse the amoeba[214]. There are also bacterial endosymbionts capable of protecting

amoebae from being infected by giant viruses, such as *Parachlamydia acanthamoebae*, which manages to suppress the viral replication of sympatric Viennavirus, APMV and Tupanvirus by inhibiting the maturation of the VF, which results ultimately in the survival of the infected amoebae[215]. Such complex systems are also present in other protists, such as *Cryptomonas* sp. SAG25.80, which is a quadripartite system[216]. A single cell harbors a phage (MAnkyphage) and two bacterial endosymbionts (*Grellia numerosa and Megaira polyxenophila*), and a complex community of organelles and selfish elements[216]. As discussed above, protist organisms have developed various strategies throughout their evolutionary history to survive unfavorable environmental conditions and escape viral infection. A graphical representation of the main strategies discussed is shown by Fig. 3.

## Summary and concluding remarks
Early eukaryotes are believed to have emerged more than a billion years ago in the Proterozoic oceans[217]. Despite the ongoing efforts to characterize and explore the diversity of viruses that infect protists, few isolates have been described to date, and the majority of the known diversity was discovered through metagenomic studies. One of the most notable groups of viruses, with the higher number of viral isolates, are the representatives of the phylum *Nucleocytoviricota*, an ancient component of the eukaryotic virome[32,218]. Each of these organisms have undergone extensive co-evolution, resulting in their present-day diversity, which encompasses a broad array of eukaryotes as hosts, and different taxa of viruses as parasites[48]. Even though certain viruses of this phylum possess a highly diverse set of genes involved in different metabolic pathways, their obligated intracellular parasitic nature necessitates an intricate relationship with their hosts.

Viruses often disrupt normal cellular processes to acquire the necessary elements for viral replication, or as a result of dysbiosis originated from the infection process. The intensification of studies on viral-protists host interactions is enhancing our understanding of the molecular mechanisms underlying their interactions, shedding light on viral replication and host defense mechanisms. Following viral infection, host cells often go through significant alterations and begin to exhibit a distinct nature[49]. Some of the alterations can be observed microscopically and are considered as an indicative of viral infection. These particular alterations, known as cytopathic effects, typically comprise morphological changes that may manifest as

noticeable variations in cell shape, size and integrity. In the case of protists, it has been observed that these changes can include increased cellular motility, rounding of cells, reduced adhesion, or increased adhesion to other cells, and cell lysis[12,80–83,85–92]. Furthermore, morphological modifications are accompanied by intracellular alterations. During a viral infection, intracellular changes in protists cells include structural transformations of the cytoskeleton, degradation of chloroplasts or photosynthetic pigments (if present), recruitment of mitochondria, membranes, vesicles, etc. Even if the VF is exclusively formed in the cytoplasm and the nucleus remains intact throughout viral replication, it can still go through changes, albeit temporary, during the course of the infection[12,66,109,117,128,133,219,220]. In addition, viral infections induce functional changes in host cells through widespread genetic and metabolic reprogramming. Apart from inducing the expression of viral genes, viruses can elevate the expression of host genes associated with processes such as DNA synthesis, energy generation, and the modification of lipid, protein, and nucleic acid biosynthesis pathways. These alterations collectively contribute to the complex interplay between viruses and host cells during infection[62,117,135–138,140,144,153,154,167]. Still, deeper exploration of virus-host dynamics is much needed, and can help elucidate host factors influencing susceptibility, pathogenicity, resistance, and protists basic cytology.

As aforementioned, protists are a diverse group of eukaryotic organisms that exhibit a remarkable adaptability when exposed to harsh conditions and invasive encounters. To thrive in ever-changing environments and fend off predators and parasites, protists have evolved a repertoire of sophisticated cellular responses to stress, often including the formation of resistant life forms. Moreover, as they occupy a wide range of ecological niches, protists have a dynamic interplay with other organisms with which they form communities, including viruses. They can act as regulators of population diversity and density, as in the case of *Emiliania huxleyi* viruses, which are responsible for ending coccolithophorid blooms[195,197]. Furthermore, the interaction between viruses and protists can also be very complex, as in tripartite systems, in which the virus-host relationship involves another member which may interfere in the replication cycle of the giant virus and protect amoebae from infection[79,212,213]. Competitive interactions are one of the main driving forces that lead to the diversity and complexity of life on our planet, and the investigation into how viruses influence protists communities, and their ecological interactions can help understand how virus-driven evolution shapes the diversity, dynamics and the impacts in ecosystems, including their role in nutrient cycling and energy flow.

Many are the taxonomic groups of viruses that infect protists, and together, they are present in the majority of Earth's ecosystems. One outcome of such interactions is horizontal gene transfer. Viruses might act as agents for transferring genetic material across species boundaries, including protists, while certain species are hot spots for horizontal gene transfer (HGT) among viruses, eukaryotes, prokaryotes and mobile elements[160,221]. This process has the potential to introduce novel genes or traits into protist genomes, and potentially influence their biology, adaptation, and evolution[222]. Furthermore, it contributes to studies on the evolution of protists and its viruses, providing insights into the coevolutionary dynamics, and as these organisms often have unique genomic features with a high percentage of ORFan genes[60], the study of the viral genomes can uncover novel genes and genetic features that may be noteworthy in protist hosts. This can expand our understanding of genetic diversity, the impact of viral genes on protists' biology, ecology and evolution, as well as potential functions within the protists.

Protists and viruses play a crucial role in Earth's ecosystems. Understanding how viruses impact protist biology can provide valuable insights into several fields of science, shedding light on evolutionary biology, ecosystem dynamics, nutrient cycling, and biomedical research. However, our knowledge of unicellular organisms and microorganisms known to humankind remains anthropocentric, with little known about the true diversity of these beings. Therefore, it is important that prospecting and characterization studies of these organisms are carried out to better elucidate the complete picture. In summary, viruses can serve as valuable tools and subjects of study in the field of protist biology. Their interactions with protist hosts, unique genomes, and ecological roles can offer new information that enhances our understanding of protist biology and its broader implications for ecosystems and biotechnology.

## Reporting summary
Further information on research design is available in the Nature Portfolio Reporting Summary linked to this article.

## Data availability
No datasets were generated or analysed for this review paper. Literature cited is shown in the reference list and in the Supplementary Data File.

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

## Acknowledgements
We thank our colleagues from Laboratório de Vírus—UFMG for their technical support. We acknowledge financial support from Rede Vírus—Ministério da Ciência, Tecnologia e Inovações (MCTI), Câmara Pox—405249/2022-5. We thank Conselho Nacional de Desenvolvimento Científico e Tecnológico (CNPq), Coordenação de Aperfeiçoamento de Pessoal de Nível Superior (CAPES), grant number 88882.348380/2010-1, Fundação de Amparo à Pesquisa do estado de Minas Gerais (FAPEMIG), Programas Institutos Nacionais de Ciência e Tecnologia (INCT), grant number 406441/2022-7, chamada 58/2022, e Pró-Reitorias de Pesquisa e Pós-Graduação of UFMG, and the Centre for New Antibacterial Strategies (CANS) of the Arctic University of Norway (project ID #2520855). R.A.L.R. and J.S.A. are CNPq researchers.

## Author contributions

V.F.Q., J.M.T. and B.B.B. performed the literature review and prepared the figures and tables. R.A.L.R., G.M.F.A. and J.S.A. designed the text structure and revised the manuscript. All authors contributed to the writing.

## Competing interests

The authors declare no competing interests.
