## [Peer Review File · Communications Biology]

Reviewers' comments:

Reviewer #1 (Remarks to the Author):

This comprehensive review covers the broad topic of protist-virus interactions, where substantial advances have been made in the last two decades through discoveries of viruses and application of new techniques. As mentioned in the manuscript, viral infection of protists is now well recognized as an important process in Earth's ecosystems and as a window into the cell biology of protists. This timely work provides a concise overview of this field, summarizing common morphological, physiological, and metabolic changes of host protist cells either as the result of viral manipulation or as cellular responses to viral infection.

Major comments:

1. While the title is "protist", this manuscript has a clear focus on the amoebas *Acanthamoeba* and *Vermamoeba*. Other major protist lineages with known viruses can be potentially better represented and described in this work. For example, it seems there is not much discussion of chlorophytes (green algae) and their viruses. One interesting cell biological change in *Ostreococcus* upon viral infection is that the ammonium uptake rate is increased by the virus-encoded ammonium transporter (<https://www.pnas.org/doi/10.1073/pnas.1708097114>). In Line 128, it is mentioned that "the viruses belonging to this phylum [Nucleocytoviricota] display a broad host range within eukaryotes", but there is no description that can help the readers to appreciate how broad it is in terms of the diversity of eukaryotes. Since the topic is protists (eukaryotes that are not animals, plants and fungi) and protists contain an extremely wide range of organisms, it is worth noting how much of this diversity is covered by the known hosts of Nucleocytoviricota.

2. Although animals are not protists, many Nucleocytoviricota viruses typically infect animals, and some of them have well documented CPEs and viral replication cycles. For readers who are more familiar with animal systems, it is helpful to outline some cell biological similarities and differences between animal-virus and protist-virus interactions.

Other comments:

Line 16 & 51: The diversity of protists undoubtedly surpass that of metazoans, not just potentially or likely.

Line 39-41: The title is "protist", but here it starts with "Protozoa". These two terms are different. Maybe just start with Protista (Ernst Haeckel 1866).

Line 53 & 120: Protists are all the eukaryote lineages that are not animals, plants, and fungi. They are mostly unicellular, but in terms of the phylogenetic relationships, they should not be referred to as simply "basal eukaryotes".

Line 71-74: Many viruses also employ the endocytic pathway to enter animal cell. A recent study (<https://journals.asm.org/doi/10.1128/spectrum.04944-22>) shows that many Nucleocytoviricota viruses of animals and amoebae have homologous entry-fusion proteins that might function in cell entry. It would be interesting to note if it is a relatively common route for cell entry.

Line 132: "protists containing chloroplasts" ==> algae (photosynthetic protists)

Line 182: It is unclear to me what the "other types" here refer to.

Line 194: "CPE effects" ==> "CPEs"

Line 418: "calcium promoting physical interactions between host receptors and viruses" Is it referring to something general or something specific to *E. huxleyi* and its virus? Please provide a citation.

Line 446-447: "The first external layer is called ectocyst, and the second is the endocyst, which is an internal fibrillar layer twice as thick as the endocyst." Maybe it is the "ectocyst" that is twice as thick as endocyst?

Line 477: What is a metabolic apparatus? Of viruses?

Chuan Ku

Reviewer #2 (Remarks to the Author):

Overview and general recommendation:

The manuscript from Fulgencio Queiroz et al. is a review on protist cell alterations caused by viral infections. First, I want to state that I am not an expert on viruses, nor amoeba cell biology. I enjoyed reading the manuscript and learned a lot, however, the manuscript should be improved to be fully convincing. I have read recent works on the virology of protists focused on molecular data, but not on the changes in the infected cells (<https://doi.org/10.1038/s41564-023-01378-y>, <https://doi.org/10.1038/s41564-021-01026-3>). As an evolutionary protistologist, I think this review is of high interest. For a non-specialized general reader, I think some more details on the cell alterations could help to increase the interest for this topic. As a review, I think it could be more completed, in terms of details of the accumulated knowledge, but also for current and future prospects in the field. I would like to have some details on the molecular bases for these cell alterations, and not just a list of altered functions.

Nevertheless, I think the article could be ready to publish after a some revisions.

General comments:

- Overall, the names of viruses should be in italics, right?
- Obviously, I have got interested in this “bunch” concept. What is so special about this type of cell aggregation, that needs a special name? I think the authors should dedicate a couple of lines explaining this concept.
- I am not a native English speaker, but I think I spotted some grammatical mistakes; or at least sentences or expressions that distracted me from the continuous reading (e.g., professional phagocytes, fend off, etc.). Overall, I perceived a tone of informality in the terminology and lack of details, although this may well be a personal preference, more than a formal critique.
- Viruses and protist have a long evolutionary story, why not starting with a phylogenetic-like figure, summarizing which lineages have been studied, how many viruses are known for each lineage, add some numbers, etc.? I mention this because the conclusion seems more like another introduction. At the same time, the intro does not focus on protist-host interactions, it mentioned tangential aspects, such the diversity of protist species, and details about amoebae, but not to protists in general. I understand that most studies on protist-virus interactions has been done in amoebae, but the authors should make a better effort to discern if this is a review on amoebae or protists in general.

Detailed comments:

L47. Among non-multicellular organisms?

L50-52. I think it is difficult to calculate the true diversity of species, could the authors give some more details on how this number has been obtained?

L56. Here the plural of amoeba is amoebae, but later in the manuscript is amoebas, should the authors choose one of the two forms?

L60-61. “, giving these 61 beings a ubiquitous character” is redundant, and it is already a long sentence.

L71. Who are these microorganisms? The giant viruses? Please avoid using this, these, those between sentences that can create confusion.

L73. Again, which strategy?

L78. What's the meaning of "unraveling of mechanisms".

L84-85. Since you start talking about amoebae, I would say: it is fundamental to study the basic aspects of amoebae cell biology in relation with virus infections.

L98-100. This sentence needs citation.

L141. A dot is missing.

L145. Section title in bold?

L264-268. Long sentence, difficult to understand. Please rewrite.

L275. Would it be Other protists' morphology and behavior?

L292. *Chaetoceros* in italics.

L323-325. I cannot understand this sentence, please rewrite.

L328. What does it mean, "differentially impacted genes"?

L339 and 341. What early and later in infection mean? The original authors might have taken the samples at specific time points, right? See the next paragraph for *Aureococcus*.

L367. Suggested instead of suggesting?

L375-377. This is a good example for why I think this review could be improved. I think this last paragraph should contain a summarized, integrative vision of differential gene expression in the various protists. I would have appreciated a figure or summary table for this section.

Section "Cellular response to extrinsic factors and infection". I think extra titles dividing into subsections per species would improve readability. Also, I would have enjoyed a figure summarizing the studies on the 3 species, which would be also nice for comparison; instead of the box, which could have been just part of the text of the same section.

L441-452. Is such a detailed explanation of amoebal cyst structure needed? It looks out of context.

Box 1, first row: which amoeba?

L474. First time a virus name is in italics in the whole manuscript.

L481. Is it in the conclusions where this virocell concept should be first mentioned? This is a clear example of why I think this manuscript could be improved by working on it a bit more.

L493-495. Again, this VF explanation could have been in the introduction to help the reader clarify the gray area in figure 1.

L500. Seriously? As far as I understood, a citation is needed to back a statement, if your statement is

so general that is backed by over 30 studies, just pick the one or two more relevant.

L519-520. As a pacifist, I try to remove belligerent anthropocentric words from my scientific texts (attack, invasion, etc.). As a reviewer, I tend to refrain myself from mentioning this, but this sentence is just plain wrong, fighting for supremacy is not a biologic concept. Please back it up with citations, or better, just remove it.

Figure 1. I have few suggestions for the authors to improve it:

Make the separating dashed lines between viruses' alterations narrower and gray in order to remove attention from the rest of the images.

Put the names of the viruses at the periphery of the image.

Some red words are difficult to read, increase font if possible.

Add what are the different drawings in a legend, I understand the different particles are the viruses, but why they have distinct colors? Removing some particles or reducing their size allows increasing the size of the font in red. Also, is the gray area, corresponding to VFs (I guess) is always a specific cell compartment? Please make an effort to increase the readability and comprehension of the main figure.

Might be impossible, but having microscopic images of some of these cell alterations would be great for the review.

Reviewer #3 (Remarks to the Author):

This ambitious review summarizes work about (giant) viruses infecting protists from just over 20 years. The manuscript focuses on the effects of viral infections on protist host cells. As such, the content overlaps partially with current reviews (co-authored by authors of the present manuscript) about giant viruses in amoeba (<https://pubmed.ncbi.nlm.nih.gov/35655338/>) and about cellular functions altered during virus infections in protists (<https://pubmed.ncbi.nlm.nih.gov/37740576/>).

Perhaps as a consequence of the large body of work and the highly diverse groups of viruses and protists covered here, some of the chapters remain slightly superficial. Also perhaps owing to the diversity and complexity of the study subjects, the review rather represents an extended list of findings than a synthesis leading to new insights or perspectives. The review, however, still provides a good starting point for anyone interested in literature about giant viruses and the consequences of giant virus infections in protists.

I have just a few, mainly editorial comments.

1. The manuscript summarizes the effects of virus infection and replication on host cell structure, physiology, gene expression, and behavior. The current title is thus misleading as the review does not provide any new insights into the cellular biology of protists based on studying viruses. A more appropriate title would be something like "The consequences of giant virus infection on protists".

2. The abstract announces that the review aimed to "explore intricate natural interactions that together these organisms carry out in ecosystems." However, insights into virus-protist interactions under natural conditions are not discussed, and the review almost exclusively summarizes work carried out under laboratory conditions. I was also not able to identify any ecosystem-level perspective in the manuscript. This statement should thus be omitted.

3. The authors start with describing microscopically visible host changes triggered by virus infections as cytopathic effects. An alternative view would be to think of the infected hosts as virocells (which is indeed mentioned in a single sentence in the last manuscript section). It would be helpful to introduce this concept and alternative, widely discussed perspective on viruses and virus-host interactions early in the manuscript text.

4. Line 58: Amoeba interact with a "plethora of microorganisms". Yet most of the references cited here are about amoeba-virus interactions. To better reflect the large body of work available about bacteria associated with amoeba, consider citing additional reviews, e.g.

<https://pubmed.ncbi.nlm.nih.gov/31049565/>

<https://pubmed.ncbi.nlm.nih.gov/15537084/>

5. Line 61: Please note that free-living amoeba represent a diverse and polyphyletic group. They are not limited to the Amoebozoa but also found in other protist lineages, e.g. Naegleria species and many other amoeba in the Heterolobosa.

6. The section entitled "Virology is an integrative science" is somewhat generic and distracts from the focus of the present review. Could be omitted to enhance reading flow.

7. Figure 1: Some of the text is too small and hardly readable, e.g. the text box in the center of panel A. The abbreviation P. should be explained (in the figure legend).

Queiroz et al rebuttal letter to the reviewers' comments

Reviewer #1 (Remarks to the Author):

This comprehensive review covers the broad topic of protist-virus interactions, where substantial advances have been made in the last two decades through discoveries of viruses and application of new techniques. As mentioned in the manuscript, viral infection of protists is now well recognized as an important process in Earth's ecosystems and as a window into the cell biology of protists. This timely work provides a concise overview of this field, summarizing common morphological, physiological, and metabolic changes of host protist cells either as the result of viral manipulation or as cellular responses to viral infection.

Answer 1: We thank you for your time and comments. Note that during the process of revising the manuscript we have expanded the information shown and prepared additional figures and tables. We will answer your comments one by one below.

Major comments:

1. While the title is "protist", this manuscript has a clear focus on the amoebas *Acanthamoeba* and *Vermamoeba*. Other major protist lineages with known viruses can be potentially better represented and described in this work. For example, it seems there is not much discussion of chlorophytes (green algae) and their viruses. One interesting cell biological change in *Ostreococcus* upon viral infection is that the ammonium uptake rate is increased by the virus-encoded ammonium transporter (<https://www.pnas.org/doi/10.1073/pnas.1708097114>). In Line 128, it is mentioned that "the viruses belonging to this phylum [Nucleocytoviricota] display a broad host range within eukaryotes", but there is no description that can help the readers to appreciate how broad it is in terms of the diversity of eukaryotes. Since the topic is protists (eukaryotes that are not animals, plants and fungi) and protists contain an extremely wide range of organisms, it is worth noting how much of this diversity is covered by the known hosts of Nucleocytoviricota.

Answer 2: Dear reviewer, this review was originally written covering most of the papers found on NCBI when searching for the terms virus and protists/protozoa. Following your comment we performed a deeper search effort using other platforms than NCBI, such as google scholar, to attempt to add more details of other protists hosts to the manuscript. This led to an increase in other papers to cite regarding other protist groups, expanding the text regarding morphological changes and the text regarding metabolic changes in protists other than amoebas. This expanded literature review is documented in the supplementary tables.

However, the current literature does indeed have a bias towards amoeba viruses. Green algae viruses were not addressed in the main text since they were considered part of the sub-kingdom *Viridiplantae*, a clade that also comprises land plants. A comprehensive view of the viruses of unicellular eukaryotes host range is better shown in the new Figure1 presenting the known viral groups and the number of virus classes that infect each protist supergroup.

2. Although animals are not protists, many Nucleocytoviricota viruses typically infect animals, and some of them have well documented CPEs and viral replication cycles. For readers who are more familiar with animal systems, it is helpful to outline some cell biological similarities and differences between animal-virus and protist-virus interactions.

Answer 3: Thank you for this observation. Indeed, at a cellular level, many conserved processes of these viruses can be observed even in the many different hosts that the representatives of this phylum have. The consequences of viral infections are, many times, detailed reported in animals, especially for viruses that have clinical relevance or economic impact. Even a brief description and comparison topic would become too large for this manuscript, which is already over the recommended word size. With this in mind, and considering that the cellular alterations shown in red in our Figure 2 may already be enough for a reader with experience in animal systems to make comparisons, we decided to keep our focus on only protist hosts.

Other comments:

Line 16 & 51: The diversity of protists undoubtedly surpasses that of metazoans, not just potentially or likely.

Answer 4: Thank you for this observation. We have modified the text in both instances to make this affirmation clearer.

Line 39-41: The title is "protist", but here it starts with "Protozoa". These two terms are different. Maybe just start with Protista (Ernst Haeckel 1866).

Answer 5: We understand that these are different terms. After considering the options and re-checking the literature we chose to keep the first formalized name of the taxon and decided to unravel what was considered part of the taxon, at the time, to avoid misleading interpretations. Small changes were made to this paragraph to make the term use decision clearer.

Line 53 & 120: Protists are all the eukaryote lineages that are not animals, plants, and fungi. They are mostly unicellular, but in terms of the phylogenetic relationships, they should not be referred to as simply "basal eukaryotes".

Answer 6: Thank you for this observation. We have modified the text in both instances.

Line 71-74: Many viruses also employ the endocytic pathway to enter animal cells. A recent study (<https://journals.asm.org/doi/10.1128/spectrum.04944-22>) shows that many Nucleocytoviricota viruses of animals and amoebae have homologous entry-fusion proteins that might function in cell entry. It would be interesting to note if it is a relatively common route for cell entry.

Answer 7: Thank you for the reference suggestion. As these viruses have different strategies to enter the host cell, we modified this paragraph and added more references that better address the viral entry step for readers eager to know more on this topic.

Line 132: "protists containing chloroplasts" ==> algae (photosynthetic protists)

Answer 8: Thank you for this observation. During the text revision this line was removed from the text.

Line 182: It is unclear to me what the "other types" here refer to.

Answer 9: Thank you for this observation. We have modified the text to make clearer that we are referring to other types of host cells.

Line 194: "CPE effects" ==> "CPEs"

Answer 10: Thank you for this observation. It was corrected.

Line 418: "calcium promoting physical interactions between host receptors and viruses" Is it referring to something general or something specific to *E. huxleyi* and its virus? Please provide a citation.

Answer 11: Thank you for the comment, the way it was written in the text was indeed confusing. Different viruses modify the calcium signaling pathway to facilitate their entry into the host cell. The text has been modified and additional citations have been provided.

Line 446-447: "The first external layer is called ectocyst, and the second is the endocyst, which is an internal fibrillar layer twice as thick as the endocyst." Maybe it is the "ectocyst" that is twice as thick as endocyst?

Answer 12: Thank you for the observation. We realized that this text part was confusing, in part due to our generalization on the theme. The morphology of cyst wall varies among species and has been used as basis of free-living amoebae identification, due to their peculiarities in each group. To avoid confusion to the readers we kept the discussion about cysts and the 'Cheshire Cat' theory while removing details about the cyst structure, which are not relevant for this discussion.

Line 477: What is a metabolic apparatus? Of viruses?

Answer 13: Thank you for this observation. 'Metabolic apparatus' was used as a virology jargon that indeed can lead to confusion. By it we refer to viral genes which are likely related to metabolic pathways. We removed the jargon and added a more detailed description.

Reviewer #2 (Remarks to the Author):

Overview and general recommendation:

The manuscript from Fulgencio Queiroz et al. is a review on protist cell alterations caused by viral infections. First, I want to state that I am not an expert on viruses, nor amoeba cell biology. I enjoyed reading the manuscript and learned a lot, however, the manuscript should be improved to be fully convincing. I have read recent works on the virology of protists focused on molecular data, but not on the changes in the infected cells (<https://doi.org/10.1038/s41564-023-01378-y>, <https://doi.org/10.1038/s41564-021-01026-3>). As an evolutionary protistologist, I think this review is of high interest. For a non-specialized general reader, I think some more details on the cell alterations could help to increase the interest for this topic. As a review, I think it could be more completed, in

terms of details of the accumulated knowledge, but also for current and future prospects in the field. I would like to have some details on the molecular bases for these cell alterations, and not just a list of altered functions.

Nevertheless, I think the article could be ready to publish after a some revisions.

Answer 14: We thank you for your time and comments. Note that during the process of revising the manuscript we have expanded the information shown and prepared additional figures and tables, besides including more information on metabolic changes and gene regulation as a result of viral infections. We will answer your comments one by one below.

General comments:

- Overall, the names of viruses should be in italics, right?

Answer 15: Dear reviewer, in this text we are following the guidelines of the International Committee on Taxonomy of Viruses, which can be verified at <https://ictv.global/faq/names> and also in Zerbini et al 2022 - 10.1007/s00705-021-05323-4. According to the guidelines, the name of the viral entity should never be italicized and should only begin with capital when these words are proper nouns. Furthermore, a collective name is neither italicized nor capitalized. In contrast, virus species and other taxonomic categories of viruses, even above the level of genus, are written in italics.

- Obviously, I have got interested in this “bunch” concept. What is so special about this type of cell aggregation, that needs a special name? I think the authors should dedicate a couple of lines explaining this concept.

Answer 16: The ‘bunch’ phenomenon has been associated as a response to certain viral infections. It is believed to help in increasing virus dissemination by keeping cells close together. Given the importance and ‘uniqueness’ of this phenomenon, further details of this concept were added to the manuscript.

- I am not a native English speaker, but I think I spotted some grammatical mistakes; or at least sentences or expressions that distracted me from the continuous reading (e.g., professional phagocytes, fend off, etc.). Overall, I perceived a tone of informality in the terminology and lack of details, although this may well be a personal preference, more than a formal critique.

Answer 17: Thank you for the observation. We carefully screened the text for typos, grammatical mistakes, jargons and unusual expressions, changing them whenever possible.

- Viruses and protist have a long evolutionary story, why not starting with a phylogenetic-like figure, summarizing which lineages have been studied, how many viruses are known for each lineage, add some numbers, etc.? I mention this because the conclusion seems more like another introduction. At the same time, the intro does not focus on protist-host interactions, it mentioned tangential aspects, such the diversity of protist species, and details about amoebae, but not to protists in general. I understand that most studies on protist-virus interactions has been done in amoebae, but the authors should make a better effort to discern if this is a review on amoebae or protists in general.

Answer 18: Thank you for your comment and excellent idea for a figure. As already mentioned to reviewer #1, a comprehensive view of the viruses of unicellular eukaryotes host range is better shown in the new Figure1 presenting the known viral groups and the number of virus classes that infect each protist supergroup. We added more details about of other protists host to the manuscript, but the current literature does indeed have a bias towards amoeba viruses. Regarding your comments about the conclusions and introduction, changes were made to the introduction, and current and future prospects were further explored in the conclusion.

Detailed comments:

L47. Among non-multicellular organisms?

Answer 19: Thank you for the observation. After rewriting this part to accommodate the reviewer comments this line was removed from the text.

L50-52. I think it is difficult to calculate the true diversity of species, could the authors give some more details on how this number has been obtained?

Answer 20: This number was obtained from reference number 7 (Brusca, R., Shuster, S. & Moore, W. Invertebrates), considered one of the main references of the field. We went deeper into it to explain how this number was calculated and discovered that the authors did not provide further description of how the number was obtained. To avoid any potential mistake or misconception we decided to remove the number from the manuscript, changing the exact “16%” mention to “a small portion” instead.

L56. Here the plural of amoeba is amoebae, but later in the manuscript is amoebas, should the authors choose one of the two forms?

Answer 21: Thank you for the observation. We decided to use “amoebae” throughout the text.

L60-61. “, giving these 61 beings a ubiquitous character” is redundant, and it is already a long sentence.

Answer 22: Thank you for the observation. The last part of the sentence was removed from the text.

L71. Who are these microorganisms? The giant viruses? Please avoid using this, these, those between sentences that can create confusion.

Answer 23: In this sentence we added ‘viral particles’ to be more specific. While checking the text for English language corrections, as mentioned above, we aimed at replacing other demonstratives throughout the text.

L73. Again, which strategy?

Answer 24: We specified the strategy we refer to in this sentence.

L78. What's the meaning of "unraveling of mechanisms".

Answer 25: The “unraveling of mechanisms” refers to the process of understanding and elucidating the details underlying, in this case, the resistance and pathogenicity of microorganisms.

L84-85. Since you start talking about amoebae, I would say: it is fundamental to study the basic aspects of amoebae cell biology in relation with virus infections.

Answer 26: It is undeniable that the manuscript brings more information on the consequences of viral infection in amoebae, but this reflects what is available in the literature. In this work, we tried to address the changes in all protists, so we chose to keep it the way it was written, and added more details of other protists hosts in the manuscript.

L98-100. This sentence needs citation.

Answer 27: We have provided citations to this part of the text.

L141. A dot is missing.

Answer 28: Thank you for the observation. The dot was added.

L145. Section title in bold?

Answer 29: Thank you for the comment. Considering the comments of other reviewer, this sentence is no more a topic title and thus does not need to be in bold.

L264-268. Long sentence, difficult to understand. Please rewrite.

Answer 30: Thank you for the observation. The sentence has been separated into two phrases to become less confusing.

L275. Would it be Other protists' morphology and behavior?

Answer 31: Thank you for the observation. It was corrected.

L292. Chaetoceros in italics.

Answer 32: Thank you for the observation. It was corrected.

L323-325. I cannot understand this sentence, please rewrite.

Answer 33: Thank you for the observation. The sentence has been rewritten to become less confusing.

L328. What does it mean, “differentially impacted genes”?

Answer 34: We meant genes which had their expression levels affected. We changed the terminology for clarification.

L339 and 341. What early and later in infection mean? The original authors might have taken the samples at specific time points, right? See the next paragraph for Aureococcus.

Answer 35: The time points used by the authors were specified in the text.

L367. Suggested instead of suggesting?

Answer 36: Thank you for the observation. It has been changed to ‘suggested’.

L375-377. This is a good example for why I think this review could be improved. I think this last paragraph should contain a summarized, integrative vision of differential gene expression in the various protists. I would have appreciated a figure or summary table for this section.

Answer 37: Thank you for your point. We added a table (Table1 of the manuscript) compiling the studies that provided information on how many proteins or transcripts were regulated. Unfortunately, several papers just discussed the main results of transcriptomic and proteomic research on text, and/or used other approaches to study metabolic differences upon infection, without providing enough information for comparing with the other studies. Regarding an integrative vision, although it is difficult to summarize the changes considering how unique these group of protists are and how different methodological approaches were in each study, we made some comparison while describing the metabolic changes. There is also an overview at the summary and concluding remarks.

Section “Cellular response to extrinsic factors and infection”. I think extra titles dividing into subsections per species would improve readability. Also, I would have enjoyed a figure summarizing the studies on the 3 species, which would be also nice for comparison; instead of the box, which could have been just part of the text of the same section.

Answer 38: Thank you for the observation. This section was divided into subsections as suggested. The box was removed, and a new figure was included as suggested (Figure 3). It is important to mention that the box was suggested by the Editor when discussing the planning of the article, and that the substitution of the box to Figure3 is subject to the editor's approval.

L441-452. Is such a detailed explanation of amoebal cyst structure needed? It looks out of context.

Answer 39: Reviewer #1 also commented on this topic. We decided to simplify this paragraph, keeping the relevant biological interaction discussion while shortening the description of cyst structure.

Box 1, first row: which amoeba?

Answer 40: Since we are compiling different viruses, the amoeba species vary. Mimiviruses do this in amoebae of the genus *Acanthamoeba*, and Tupanvirus does this in amoebae of the genus *Acanthamoeba* and also in *Vermamoeba vermiformis*. We decided not to restrict the species and leave the general term 'amoeba' instead.

L474. First time a virus name is in italics in the whole manuscript.

Answer 41: According to the guidelines mentioned in our answer to your first general comment, taxonomic ranks of viruses should always be written in italics.

L481. Is it in the conclusions where this virocell concept should be first mentioned? This is a clear example of why I think this manuscript could be improved by working on it a bit more.

Answer 42: Thank you for your comment. The virocell concept has now been introduced early on in the manuscript.

L493-495. Again, this VF explanation could have been in the introduction to help the reader clarify the gray area in figure 1.

Answer 43: Together with the virocell concept, the viral factory concept has now been introduced early on in the manuscript.

L500. Seriously? As far as I understood, a citation is needed to back a statement, if your statement is so general that is backed by over 30 studies, just pick the one or two more relevant.

Answer 44: Thank you for the observation. Originally we thought about citing all the references on our reference list that supports this affirmation, given that this is a well proved phenomenon. We have modified the referencing in this sentence and only the more relevant references were kept.

L519-520. As a pacifist, I try to remove belligerent anthropocentric words from my scientific texts (attack, invasion, etc.). As a reviewer, I tend to refrain myself from mentioning this, but this sentence is just plain wrong, fighting for supremacy is not a biologic concept. Please back it up with citations, or better, just remove it.

Answer 45: Thank you for your comment. Such terminology is often used in the virus-host interaction field and belligerent terms are well accepted and recurrent in the virology literature. We agree that the use of anthropocentric words can reflect or perpetuate anthropocentric thinking, which is a biased and precarious way of interpreting and describing nature. In accordance with your statements, we replaced our words with "competitive interactions" to avoid misleading.

Figure 1. I have few suggestions for the authors to improve it:

Make the separating dashed lines between viruses' alterations narrower and gray in order to remove attention from the rest of the images.

Put the names of the viruses at the periphery of the image.

Some red words are difficult to read, increase font if possible.

Add what are the different drawings in a legend, I understand the different particles are the viruses, but why they have distinct colors? Removing some particles or reducing their size allows increasing the size of the font in red. Also, is the gray area, corresponding to VFs (I guess) is always a specific cell compartment? Please make an effort to increase the readability and comprehension of the main figure.

Might be impossible, but having microscopic images of some of these cell alterations would be great for the review.

Answer 46: Thank you for your suggestions. Note that this figure now became Figure2 in the manuscript. We agree that the proposed changes improve the quality of the figure, so we followed your suggestions for the figure structure, besides improving it with additional information found during the review process. The only unfeasible change was to include real microscopic images: these are scattered in the literature and would create additional delays (besides possible issues) regarding reprint permissions.

Major changes include a side column with every virus particle and its respective names, so that it can give the reader a better idea about how many different viruses can trigger CPEs, and a miniature overview of the amoeba cell indicating the cell, nucleus and viral factory. The colours used are just to represent different viruses.

Minor changes in Figure 2A were: melbournevirus was added to the top left ; cedratvirus location in the figure changed ; viruses which only trigger VFs were omitted since VF formation is more related to the virus than to the protist; amoeba aggregation and the viruses Port-miou virus, yoshiyabu, oosagami, shinsuna, Kurlavirus, saibaishi, koaze were added ; amoeba aggregation was also associated to Noumeavirus and Lausannevirus. Hokutovirus was moved from Figure 2A to Figure 2B since it has also been associated to cell rounding.

Minor changes in Figure 2B were: Hokutovirus was added to Figure 2B and associated to amoeba aggregation ; megavirus, viennavirus and samba virus were associated to cell rounding.

Reviewer #3 (Remarks to the Author):

This ambitious review summarizes work about (giant) viruses infecting protists from just over 20 years. The manuscript focuses on the effects of viral infections on protist host cells. As such, the content overlaps partially with current reviews (co-authored by authors of the present manuscript) about giant viruses in amoeba (<https://pubmed.ncbi.nlm.nih.gov/35655338/>) and about cellular functions altered during virus infections in protists (<https://pubmed.ncbi.nlm.nih.gov/37740576/>).

Perhaps as a consequence of the large body of work and the highly diverse groups of viruses and protists covered here, some of the chapters remain slightly superficial. Also perhaps owing to the

diversity and complexity of the study subjects, the review rather represents an extended list of findings than a synthesis leading to new insights or perspectives. The review, however, still provides a good starting point for anyone interested in literature about giant viruses and the consequences of giant virus infections in protists.

Answer 47: We thank you for your time and comments. Note that during the process of revising the manuscript we have expanded the information shown and prepared additional figures and tables, hopefully giving depth to the review text and improving its uniqueness and relevance as a summary of the topic. We will answer your comments one by one below.

I have just a few, mainly editorial comments.

1. The manuscript summarizes the effects of virus infection and replication on host cell structure, physiology, gene expression, and behavior. The current title is thus misleading as the review does not provide any new insights into the cellular biology of protists based on studying viruses. A more appropriate title would be something like “The consequences of giant virus infection on protists”.

Answer 48: Thank you for your observation. Changing the title is a major change and we decided to follow your suggestion for agreeing with your reasoning.

2. The abstract announces that the review aimed to “explore intricate natural interactions that together these organisms carry out in ecosystems.” However, insights into virus-protist interactions under natural conditions are not discussed, and the review almost exclusively summarizes work carried out under laboratory conditions. I was also not able to identify any ecosystem-level perspective in the manuscript. This statement should thus be omitted.

Answer 49: Thank you for your observation. We have modified our abstract to accommodate the idea that our references come mostly from laboratory experimentation and not field observations.

3. The authors start with describing microscopically visible host changes triggered by virus infections as cytopathic effects. An alternative view would be to think of the infected hosts as virocells (which is indeed mentioned in a single sentence in the last manuscript section). It would be helpful to introduce this concept and alternative, widely discussed perspective on viruses and virus-host interactions early in the manuscript text.

Answer 50: Thank you for your comment. The virocell concept has now been introduced early in the manuscript to properly acknowledge it and catch the reader attention to it.

4. Line 58: Amoeba interact with a “plethora of microorganisms”. Yet most of the references cited here are about amoeba-virus interactions. To better reflect the large body of work available about bacteria associated with amoeba, consider citing additional reviews, e.g.

<https://pubmed.ncbi.nlm.nih.gov/31049565/>

<https://pubmed.ncbi.nlm.nih.gov/15537084/>

Answer 51: Thank you for the comment. These references were added to support the idea that amoebae indeed interact with a plethora of other organisms besides viruses.

5. Line 61: Please note that free-living amoeba represent a diverse and polyphyletic group. They are not limited to the Amoebozoa but also found in other protist lineages, e.g. Naegleria species and many other amoeba in the Heterolobosa.

Answer 52: Thank you for the comment. We changed 'Amoebozoa' to 'Amoebae' to avoid the impression of them being monophyletic.

6. The section entitled "Virology is an integrative science" is somewhat generic and distracts from the focus of the present review. Could be omitted to enhance reading flow.

Answer 53: Thank you for your comment. The section has now been adjusted into the final paragraph of the "viruses are unique tools to study protists biology" section.

7. Figure 1: Some of the text is too small and hardly readable, e.g. the text box in the center of panel A. The abbreviation P. should be explained (in the figure legend).

Answer 54: Thank you for your comment. Note that this is Figure2 now in the revised manuscript. This figure has been extensively modified following reviewer #2 comments and we believe it is better from many perspectives now. The full name of all viruses shown is now written in the right-hand side of the figure.

REVIEWERS' COMMENTS:

Reviewer #1 (Remarks to the Author):

The authors have fully addressed the concerns and suggestions in the previous review report. I have no further comments to add.

Reviewer #2 (Remarks to the Author):

Title: The consequences of viral infection on protists

COMMSBIO-23-4092A

Overview and general recommendation:

The revised manuscript from Fulgencio Queiroz et al. is a review on protist cell alterations caused by viral infections. It has been substantially improved since the first submission, and the authors have addressed the reviewers comments appropriately. As a suggestion, I think the response letter should contain the corrected sentence, so it is easier for the reviewer to be sure that the amendment is done properly (line numbers do not mean anything after such a revised manuscript). The expanded reported protists, the new figures and table really make this review a good encyclopaedic piece. I would have loved that the authors would enter to speculate in how to address the questions of the origin and evolution of these protist-virus interactions, but in my opinion this manuscript is ready for publication.

Minor comments:

L225: If this paragraph is to talk about flagellates, I would start like: Regarding flagellates, the protists Giardia...

L267: 1st time citing HPI, must be described.

L317: particle in singular?

L566: ORFan genes? Should it be orphan genes, ORFs?

The two last paragraphs are a bit redundant, I suggest to summarize it in one.

Figure 1 legend. The acronyms from the figure should be explained.

L602: Amoebae

Reviewer #3 (Remarks to the Author):

All my comments on the original manuscript have been addressed adequately in this revised version.

Queiroz et al rebuttal letter to the second round of revision

REVIEWERS' COMMENTS:

Reviewer #1 (Remarks to the Author):

The authors have fully addressed the concerns and suggestions in the previous review report. I have no further comments to add.

Answer 1: We thank you for your time and for the positive response to our manuscript.

Reviewer #2 (Remarks to the Author):

Title: The consequences of viral infection on protists

COMMSBIO-23-4092A

Overview and general recommendation:

The revised manuscript from Fulgencio Queiroz et al. is a review on protist cell alterations caused by viral infections. It has been substantially improved since the first submission, and the authors have addressed the reviewers comments appropriately. As a suggestion, I think the response letter should contain the corrected sentence, so it is easier for the reviewer to be sure that the amendment is done properly (line numbers do not mean anything after such a revised manuscript). The expanded reported protists, the new figures and table really make this review a good encyclopaedic piece. I would have loved that the authors would enter to speculate in how to address the questions of the origin and evolution of these protist-virus interactions, but in my opinion this manuscript is ready for publication.

Answer 2: We thank you for your time and for the positive response to our manuscript.

Minor comments:

L225: If this paragraph is to talk about flagellates, I would start like: Regarding flagellates, the protists Giardia...

L267: 1st time citing HPI, must be described.

L317: particle in singular?

L566: ORFan genes? Should it be orphan genes, ORFs?

The two last paragraphs are a bit redundant, I suggest to summarize it in one.

Figure 1 legend. The acronyms from the figure should be explained.

L602: Amoebae

Answer 3: All your minor comments above have been addressed.

- Lines 225, 267 and 602 were modified according to your comments.
- We kept particles in plural on line 317 since we are referring to the whole viral progeny and not a single particle in this sentence.
- In line 566 we kept ORFan genes to keep the same designation used in the reference 60 cited.
- We tried to reduce and summarize the last two paragraphs but found it tricky because either it becomes too convoluted into one paragraph or we need to reduce information provided. Our goal with these last two paragraphs was to have the second to last closing the summary, and the last as a broad concluding remark to finish the text. I hope you don't mind if we keep the structure as it was.
- We have updated the figure 1 legend.

Reviewer #3 (Remarks to the Author):

All my comments on the original manuscript have been addressed adequately in this revised version.

Answer 4: We thank you for your time and for the positive response to our manuscript.